# Sibling similarity can reveal key insights into genetic architecture

Tade Souaiaia[1]*, Hei Man Wu[2], Clive Hoggart[2]*[†], Paul F O'Reilly[2]*[†]

[1]Department of Cellular Biology, SUNY Downstate Health Sciences, Brooklyn, United States; [2]Department of Genetics and Genomic Sciences, Icahn School of Medicine, Mount Sinai, New York, United States

## eLife assessment

The authors present a **solid** statistical framework for using sibling phenotype data to assess whether there is evidence for de novo or rare variants causing extreme trait values. Their **valuable** method is promising and will be of interest to researchers studying complex trait genetics.

***For correspondence:**
tade.souaiaia@gmail.com (TS);
clive.hoggart@mssm.edu (CH);
paul.oreilly@mssm.edu (PFO'R)

[†]These authors contributed equally to this work

**Competing interest:** The authors declare that no competing interests exist.

**Abstract** The use of siblings to infer the factors influencing complex traits has been a cornerstone of quantitative genetics. Here, we utilise siblings for a novel application: the inference of genetic architecture, specifically that relating to individuals with extreme trait values (e.g. in the top 1%). Inferring the genetic architecture most relevant to this group of individuals is important because they are at the greatest risk of disease and may be more likely to harbour rare variants of large effect due to natural selection. We develop a theoretical framework that derives expected distributions of sibling trait values based on an index sibling's trait value, estimated trait heritability, and null assumptions that include infinitesimal genetic effects and environmental factors that are either controlled for or have combined Gaussian effects. This framework is then used to develop statistical tests powered to distinguish between trait tails characterised by common polygenic architecture from those that include substantial enrichments of de novo or rare variant (Mendelian) architecture. We apply our tests to UK Biobank data here, although we note that they can be used to infer genetic architecture in any cohort or health registry that includes siblings and their trait values, since these tests do not use genetic data. We describe how our approach has the potential to help disentangle the genetic and environmental causes of extreme trait values, and to improve the design and power of future sequencing studies to detect rare variants.

## Introduction

The fields of quantitative genetics and genetic epidemiology have exploited the shared genetics and environment of siblings in many applications, notably to estimate heritability, using theory developed a century ago (*Fisher, 1919*; *Lush, 1940*), and more recently to infer a so-called 'household effect' (*Selzam et al., 2018*), which contributes to genetic risk indirectly via correlation between genetics and household environment. Here, we leverage sibling trait data to infer genetic architecture departing from polygenicity, specifically affecting the tails of the trait distribution and consistent with enrichment in rare variants of large effect.

The genetic architecture of a complex trait is typically inferred from findings of multiple independent studies: genome-wide association studies (GWAS) identifying common variants (*Yengo et al., 2022*; *Selvaraj et al., 2022*), whole-exome or whole-genome sequencing studies detecting rare variants (*Singh et al., 2022*), and family sequencing studies designed to identify de novo and rare Mendelian mutations (*Iyegbe and O'Reilly, 2022*). The relative contribution of each type of variant to trait

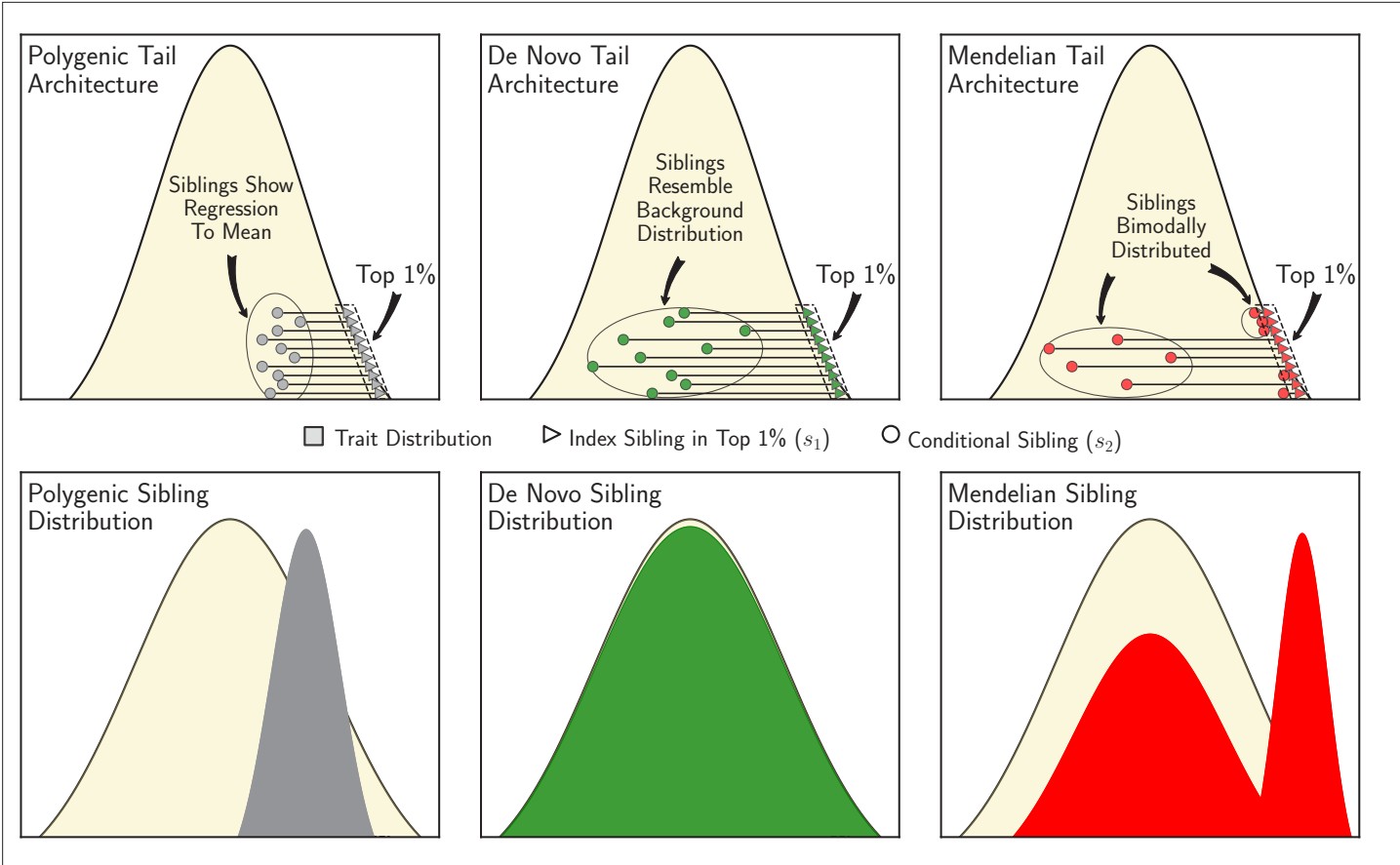

**Figure 1.** Sibling similarity under different tail architectures. Left to right: When an individual's extreme trait value (top 1%) is due to many alleles of small effect ('polygenic'), then their siblings' trait values are expected to show regression-to-the-mean (grey). When an individual's extreme trait value is due to a de novo mutation of large effect, then their siblings are expected to have trait values that correspond to the background distribution (green). When an individual's extreme trait value is the result of an inherited rare allele of large effect ('Mendelian'), then their siblings are expected to have either similarly extreme trait values or trait values that are drawn from the background distribution (red), depending on whether or not they inherited the same large effect allele.

heritability is a function of historical selection pressures on the trait in the population (*Koch et al., 2024*; *Uricchio, 2020*). If selection has recently acted to increase the average value of a trait, then the lower tail of the trait distribution may become enriched for large-effect rare variants over time because trait-reducing alleles will be subject to negative selection, and so those with large effects – most likely to 'push' individuals to the lower tail – will be reduced to low frequencies (*Corte et al., 2023*). Similarly, if the trait is subject to stabilising selection, then both tails of the trait distribution may be enriched for rare variant aetiology (*Momozawa and Mizukami, 2021*). This can result in less accurate polygenic scores in the tails of the trait distribution (*Chan et al., 2011*) and can also produce dissimilarity between siblings beyond what is expected under polygenicity. For example, studies on the intellectual ability of sibling pairs have demonstrated similarity for average intellectual ability (*Shakeshaft et al., 2015*), regression-to-the-mean for siblings at the upper tail of the distribution (*Shakeshaft et al., 2015*), and complete discordance when one sibling is at the lower extreme tail of the distribution (*Reichenberg et al., 2016*). However, no theoretical framework has been developed to formally infer genetic architecture from sibling trait data.

We introduce a novel theoretical framework that allows widely available sibling trait data in population cohorts and health registries to be leveraged to perform statistical tests that can infer complex genetic architecture in the tails of the trait distribution. These tests are powered to differentiate between polygenic, de novo, and Mendelian (i.e. rare variants of large effect) architectures (see *Figure 1*); while these are simplifications of true complex architectures, our tests allow enrichments of the different architectures to be inferred and compared across traits. This framework establishes

expectations about the trait distributions of siblings of index individuals with extreme trait values (e.g. the top 1% of the trait) according to null assumptions of polygenicity, random mating, and environmental factors that are either controlled for or have combined Gaussian effects. Critical to our framework is our derivation of the 'conditional sibling trait distribution', which describes the trait distribution for one individual given the quantile value of one or more 'index' siblings. Our statistical framework, derivation of the conditional sibling trait distribution, and simulation study allow us to develop statistical tests to infer genetic architecture from data on siblings (*Hur et al., 2013*) without the need for genetic data (see 'Model and methods'). We validate the statistical power of our tests using simulated data and apply our tests across a range of traits from the UK Biobank. We also release *sibArc*, an open-source software tool that can be used to apply our tests to sibling trait data (*sibArc: Software for Inference of Genetic Architecture, 2024*). Our novel framework can be extended to applications such as estimating heritability, characterising mating patterns, and inferring historical selection pressures.

## Model and methods

Here, we outline a framework that models the *conditional sibling trait distribution*, which describes an individual's trait distribution conditioned on an *index* sibling's trait value. In this section, we describe our derivation of the distribution for a completely polygenic trait, outline the development of statistical tests that infer enrichments of rare genetic architectures via departures from polygenicity, and describe the simulation study used to validate and benchmark our results. Full derivations that complement this high-level summary of our methodology, as well as extended results from application of our tests to UK Biobank data, are included in the Appendices.

### Conditional sibling inference framework

Assuming completely polygenic (i.e. infinitesimal) architecture, siblings of individuals with extreme trait values are expected to be less extreme. This regression-to-the-mean can be understood through the two factors (*Falconer and Mackay, 1983*; *Alberts et al., 2002*) that determine inherited genetic variation: (i) average genetic contribution of the parents (*mid-parent*) to the trait value and (ii) random genetic reassortment occurring during meiosis. Assuming that an individual presenting an upper-tail trait value does so due to *both* high mid-parent trait value *and* genetic reassortment favouring a higher value, then a sibling sharing common mid-parent trait value – but subject to independent reassortment – is likely to be less extreme. How much less extreme can be derived by first considering the conditional distribution that relates mid-parents and their offspring? In the simplest case, for a completely heritable continuous polygenic trait in a large population, the offspring trait values, $s$, are normally distributed around their mid-parent trait value, $m$, irrespective of selection and population structure (*Johnson and Barton, 2005*; *Barton et al., 2017*):

$$p(s \mid m) = \mathcal{N}\left(m, \sigma_f^2\right) \tag{1}$$

where $\sigma_f^2$ represents within-family variance, which is constant across the population. This result does not require the trait distribution across the population to be Gaussian. If we assume that the population distribution is Gaussian and there is random mating and no selection, then $\sigma_f^2 =$ half the population variance (*Bulmer, 1998*; *Johnson and Barton, 2005*; *Baselmans et al., 2021*). For a trait with heritability $h^2$, trait variance can be partitioned into genetic, $\sigma_g^2$, and environmental, $\sigma_e^2$, contributions, such that $\sigma^2 = \sigma_g^2 + \sigma_e^2$ and $h^2 = \sigma_g^2/(\sigma_g^2 + \sigma_e^2)$. Assuming mean-centred genetic and environmental trait contributions, and a trait variance of 1 (a standard normalised trait), the distribution for offspring conditioned on the mid-parent genetic trait value, $m_g$, is

$$p(s \mid m_g) = \mathcal{N}\left(m_g, \sigma_g^2/2 + \sigma_e^2\right) = \mathcal{N}\left(m_g, 1 - \frac{h^2}{2}\right) \tag{2}$$

Since $p(m_g) = \mathcal{N}(0, h^2/2)$, from Bayes' theorem, it can be shown that the mid-parent genetic trait value conditional on offspring trait value, $s$, is distributed as follows:

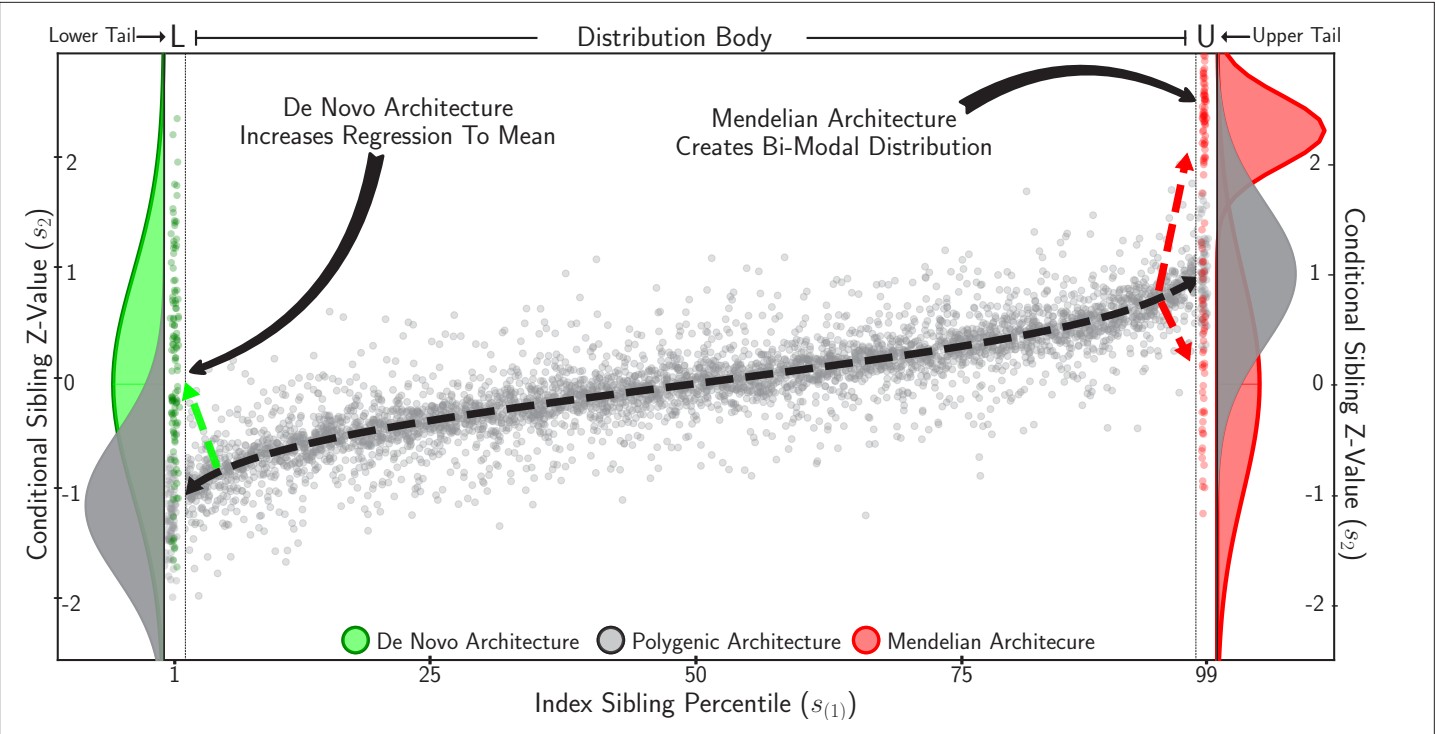

**Figure 2.** Identifying complex tail genetic architecture. Conditional sibling z-values plotted against index sibling quantiles. Grey depicts complete polygenic architecture across index sibling values. In the lower tail, an extreme scenario of de novo architecture is shown in green, resulting in sibling discordance. In the *upper tail*, extreme Mendelian architecture is shown in red, whereby siblings are half concordant and half discordant, resulting in a bimodal conditional sibling trait distribution. Statistical tests to infer each type of complex tail architecture are designed to exploit these expected trait distributions.

$$p(m_g \mid s) = \mathcal{N}\left(\frac{1}{2}sh^2, \frac{h^2}{2}\left(1 - \frac{h^2}{2}\right)\right) \tag{3}$$

From *Equations 2 and 3*, the sibling trait distribution conditional on an index sibling can be calculated as

$$
\begin{aligned}
p(s_2 \mid s_1) &= \int p(s_2 \mid m_g)p(m_g \mid s_1)\,dm_g \\
&= \mathcal{N}\left(\frac{1}{2}s_1 h^2, 1 - \frac{h^4}{4}\right)
\end{aligned}
\tag{4}
$$

A full derivation is provided in Appendix 1. This derivation via mid-parents can be generalised to cases where assumptions of random mating, Gaussian population trait distribution, and no selection do not hold. When these assumptions do hold, then the conditional sibling distribution can also be derived from the joint sibling distribution defined by the relationship matrix (*Lencz et al., 2021*). In the Appendices, we use the joint sibling distribution to derive the distribution conditional on two sibling values. From *Equation 4*, we can infer that under complete heritability the siblings of an individual with a standard normal trait value of $z$ will have a mean trait value of $\frac{z}{2}$, with variance equal to three quarters of the population variance. In Appendix 1, we further generalise the result to binary phenotypes, increasing the utility of this framework for further applications and theoretical development.

## Statistical tests for complex tail architecture

In *Figure 2*, the strategy employed to develop statistical tests for complex tail architecture is depicted. Our approach corresponds to testing for deviations from the expected conditional sibling trait distribution under the null hypothesis of complete polygenicity in trait tails (see above for other null assumptions): excess discordance is indicative of an enrichment of de novo mutations, while excess

concordance indicates an enrichment of Mendelian variants, that is, large effect variants segregating in the population. The heritability, $h^2$, required to define the null distribution, is estimated by maximising the log-likelihood of the conditional sibling trait distribution (*Equation 4*) with respect to $h^2$:

$$\log \mathcal{L} \propto -n \log \left( 1 - \frac{h^4}{4} \right) - \frac{1}{1 - \frac{h^4}{4}} \sum_{i=1}^{n} \left( s_2^{(i)} - \frac{s_1^{(i)} h^2}{2} \right)^2 \tag{5}$$

where $s_1$ and $s_2$ represent index and conditional sibling trait values, respectively, and $n$ is the number of sibling pairs. This allows $h^2$ to be estimated for given quantiles of the trait distribution by restricting sibling pair observations to those index siblings in the quantile of interest. To maximise power to detect non-polygenic architecture in the tails of the trait distribution, we estimate 'polygenic heritability', $h_p^2$, from sibling pairs for which the index sibling trait value is between the 5th and 95th percentile (labelled 'Distribution body' in *Figure 2*).

Tests for complex architecture are then performed in relation to index siblings whose trait values are in the tails of the distribution (e.g. the lower and upper 1%). Below, $A_q$ denotes the set of sibling pairs for which the index sibling is in quantile $q$, such that $s_1 > \Phi^{-1}(1 - q)$ and $s_1 < \Phi^{-1}(q)$ for the upper and lower tails, respectively, where $\Phi^{-1}$ is the inverse normal cumulative distribution function.

It should be noted that our estimate of $h^2$ in *Equation 5* assumes no effects of shared environment. *Polderman et al., 2015* found limited contribution of shared environment for most complex traits and, critically, our statistical tests are robust to shared environmental effects with consistent effects throughout the trait distribution (see 'Discussion').

## Statistical test for de novo architecture

For inference of de novo architecture in the tails of the trait distribution, we introduce a parameter, $\alpha$, to the log-likelihood defined by the conditional sibling trait distribution *Equation 5*:

$$\log \mathcal{L} \propto \frac{1}{2(1 - \frac{h^4}{4})} \sum_{i \in A_q} \left( s_2^{(i)} - \left( \frac{1}{2} s_1^{(i)} h^2 + \alpha \right) \right)^2 \tag{6}$$

Values of $\alpha > 0$ in the lower tail and $\alpha < 0$ in the upper tail indicate excess regression-to-the-mean and, thus, high sibling discordance, consistent with an enrichment of de novo mutations among the index siblings. The z-statistic of the one-sided score test for $\alpha > 0$ in the lower quantile, $q$, relative to the null of $\alpha = 0$ is (see Appendix 2 for derivation):

$$z = \frac{\sum_{i \in A_q} s_2^{(i)} - \frac{1}{2} s_1^{(i)} h^2}{\sqrt{n(1 - \frac{h^4}{4})}} \tag{7}$$

For the upper tail test of $\alpha < 0$, the above is multiplied by –1.

## Statistical test for Mendelian architecture

For inference of Mendelian architecture in the tails of the trait distribution, we compare the observed and expected tail sibling concordance, defined by the number of sibling pairs for which both siblings have trait values in the tail. For each index sibling in $A_q$, we calculate the probability that the conditional sibling is also in $A_q$, which, for the upper tail, is given by

$$P \left( s_2 > \Phi^{-1}(q) \mid s_1 \right) = 1 - \Phi \left( \frac{\Phi^{-1}(q) - \frac{s_1 h^2}{2}}{\sqrt{1 - \frac{h^4}{4}}} \right) \tag{8}$$

where $\Phi$ represents the normal cumulative distribution function. Denoting the mean of $P(s_2^{(i)} > \Phi^{-1}(q))$ across all index siblings in $A_q$ by $\pi_o$, the expected sibling concordance is $n\pi_o$, where $n$ is the number of index siblings in $A_q$. Given an observed number of concordant siblings, $r$, the z-statistic for a one-sided score test for excess concordance is (see Appendix 2 for derivation) given by

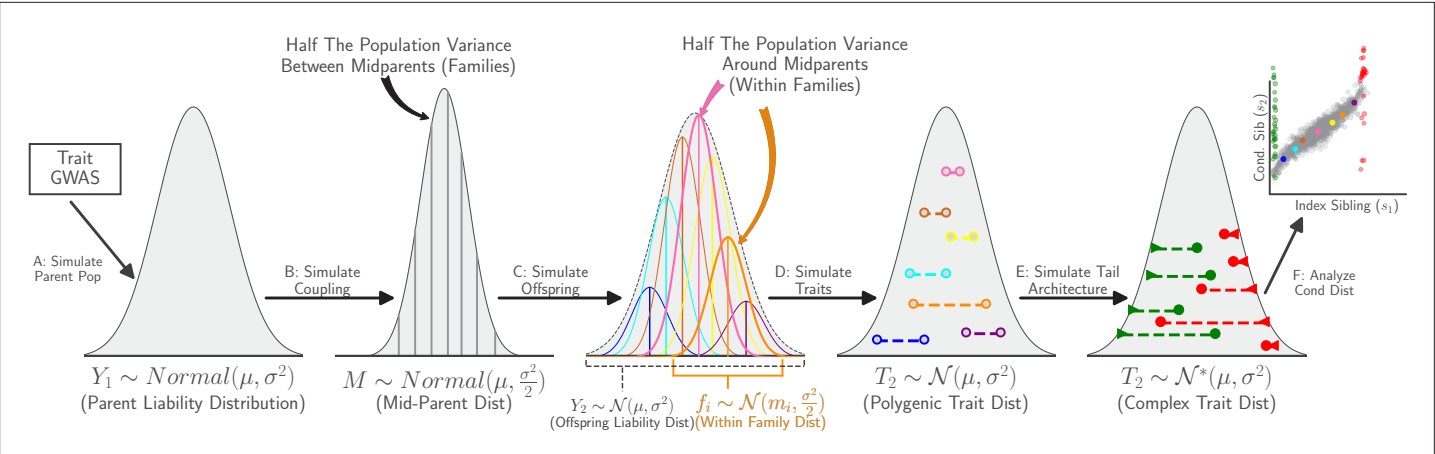

**Figure 3.** Simulation schematic. Publicly available genome-wide association studies (GWAS) allele frequency and effect size data is used to simulate parent genetic trait value (**A**). Mid-parent genetic trait value (**B**) is simulated assuming random mating. Offspring genotype and genetic trait value (**C**) is simulated assuming complete recombination. Environmental variation (**D**) is added to compare with theoretical polygenic conditional sibling distribution. De novo and Mendelian rare-variant effects are simulated (**E**) to benchmark tests for complex architecture (**F**).

$$z = \frac{r - n\pi_0}{\sqrt{n\pi_0(1 - \pi_0)}} \tag{9}$$

## Simulation of conditional sibling data

We perform simulations using publicly available GWAS data on multiple traits to validate our analytical model and benchmark our tests for complex architecture. *Figure 3* depicts the different stages of our simulation procedure. We start by simulating a 'parent population' (step A), utilising the allele frequencies and effect sizes of the first 100k SNPs from a trait GWAS to sample genotypes and subsequently trait values assuming an additive model. Next, parents are randomly paired and their genetic trait values averaged to produce mid-parent trait values (step B), and genotypes of two offspring (*Equation 1*) and corresponding genetic trait values, $G$, are calculated (step C) assuming independent reassortment of parental alleles and unlinked SNPs.

In step D, we generate offspring trait values for different degrees of heritability by adding a Gaussian environmental effect. For heritability, $h^2$, offspring trait values are given by $T = hG + E$, where $G$ is the genetic effect, standardised to have mean 0 and variance 1, and the environmental effect $E$ is drawn from a normal distribution with mean 0 and variance $(1 - h^2)$. The simulated trait has a $\mathcal{N}(0, 1)$ distribution.

In step E, we simulate the effect of complex tail architecture on the conditional sibling trait distribution. We assume that rare variants are sufficiently penetrant to move individuals into the tails of the distribution, independent of their polygenic contribution. We, thus, modify sibling trait values for individuals already in the tails (from step D) to minimise perturbation of the trait distribution. We simulate de novo tail architecture by resampling the less extreme sibling from the background distribution and simulate Mendelian tail architecture by resampling the less extreme sibling from the background distribution with probability 0.5, and from the same tail as the extreme sibling with probability 0.5.

## Application to UK Biobank data

The UK Biobank includes data from over 21,000 siblings (*Lello et al., 2023*). To apply our tests to this dataset, we began by identifying continuous traits (at least 50 unique values) with at least 5000 sibling pairs, as defined by kinship coefficient 0.18–0.35 and >0.1% SNPs with IBS0 to distinguish from parent-offspring (*Cheesman et al., 2020*; *Bycroft et al., 2018*). After removing outliers with absolute trait value >8 standard deviations from the mean, we removed all traits with absolute skew or excess kurtosis greater than 0.5 to reduce the likelihood that skewed or heavy-tailed trait distributions impact our statistical tests. The remaining traits were standardised using rank-based inverse normal transformation and adjusted for age, sex, recruitment centre, batch covariates, and the first 40 principal

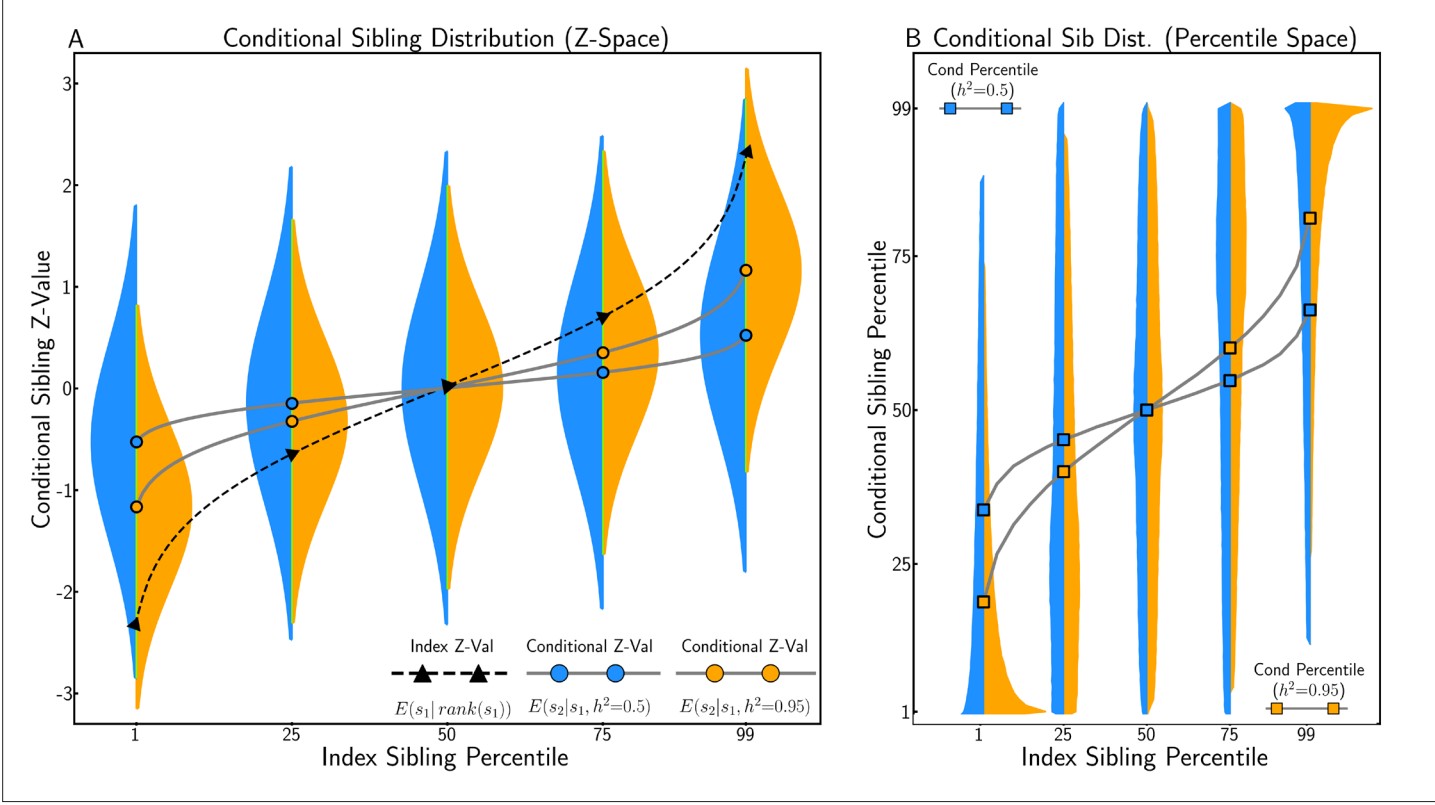

**Figure 4.** Conditional sibling trait distribution under polygenic architecture. (**A**) The conditional sibling trait distribution according to *Equation 4* for index siblings at the 1st, 25th, 50th, 75th, and 99th percentile of the standardised trait distribution, when heritability is high ($h^2 = 0.95$, in orange) and moderate ($h^2 = 0.5$, in blue). When heritability is 0.95, conditional sibling expectation is almost half of the index sibling z-score; when heritability is 0.5, the conditional sibling expectation is equal to 1/4 of the index sibling z-score. (**B**) The conditional distribution transformed into rank space. An individual whose sibling is at the 99% percentile is expected to have a trait value in the 80% percentile when heritability is high and in the 67% percentile when heritability is moderate.

components. After this, to ensure a primarily additive polygenic aetiology, we required that traits have heritability $h^2_{SNP} > 0.1$(*Ni et al., 2018*) and that no single SNP contribute more than 0.01 to $h^2$.

We applied our method to estimate heritability from sibling pairs (*Equation 5*) across the distribution and within the trait body (5th to 95th percentiles) on the remaining traits and selected the 18 traits for which both measures exceeded 30% for further analysis. For each of these 18 traits, siblings were randomly assigned index and conditional status and both trait tails were tested for departures from polygenicity using our de novo and Mendelian tests, as well as a general Kolmogorov–Smirnov test (*Marozzi, 2013*) to identify departures from the conditional sibling distribution assuming polygenicity (Appendix 2).

## Results

Here, we illustrate the conditional sibling trait distribution, validate the accuracy of our analytical model using simulation, perform power analyses for our statistical tests for complex genetic tail architecture (see 'Model and methods), and apply our tests to trait data on thousands of siblings from the UK Biobank.

### Conditional sibling trait distribution

In *Figure 4A*, the conditional sibling trait distribution (*Equation 4*) is illustrated at different index sibling trait values (ranked percentiles). For an almost entirely heritable polygenic trait (orange), siblings of individuals at the 99th percentile ($z = 2.32$) have mean z-scores approximately halfway between the

population mean and index mean (i.e. $z = 1.1$). This regression-to-the-mean is greater when trait heritability is lower (blue), assuming (as here) independent environmental risk among siblings.

In *Figure 4B*, the conditional sibling z-distribution is transformed into percentiles for interpretation in rank space. This distribution is skewed, especially at the tails, due to truncation at extreme quantiles (i.e. siblings cannot be more extreme than the top 1%). For a trait with $h^2 = 0.95$, siblings of individuals at the 99th percentile (z = 2.32) have a mean trait value at the 80th percentile. Note that this is less extreme than the result of transforming their expected z-value into percentile space ($\Phi^{-1}(1.1) = 86\%$), which is a consequence of Jensen's inequality (*Jensen, 1906*) given that the inverse cumulative distribution function of the normal is convex above zero.

We compared the theoretical conditional sibling trait distributions to those generated from simulated data (see Appendix 3) and found that irrespective of the trait used to simulate data (e.g. fluid intelligence, height) the two distributions did not differ significantly, suggesting that our analytically derived distributions are a valid model for the conditional sibling trait distribution (*Equation 4*).

## Power of statistical tests to identify complex tail architecture

Building on the theoretical framework introduced in the section 'Model and methods' and illustrated in the previous section, we develop statistical tests to identify complex architecture in the tails of the trait distribution. These tests leverage the fact that the similarity (or dissimilarity) in trait values among siblings provides information about the underlying genetic architecture (see *Figure 1*). For example, high-impact de novo mutations generate large dissimilarity between siblings when only one carries the unique mutant allele, while Mendelian variants can create excess similarity in the tails of the distribution when siblings both inherit the same mutant allele.

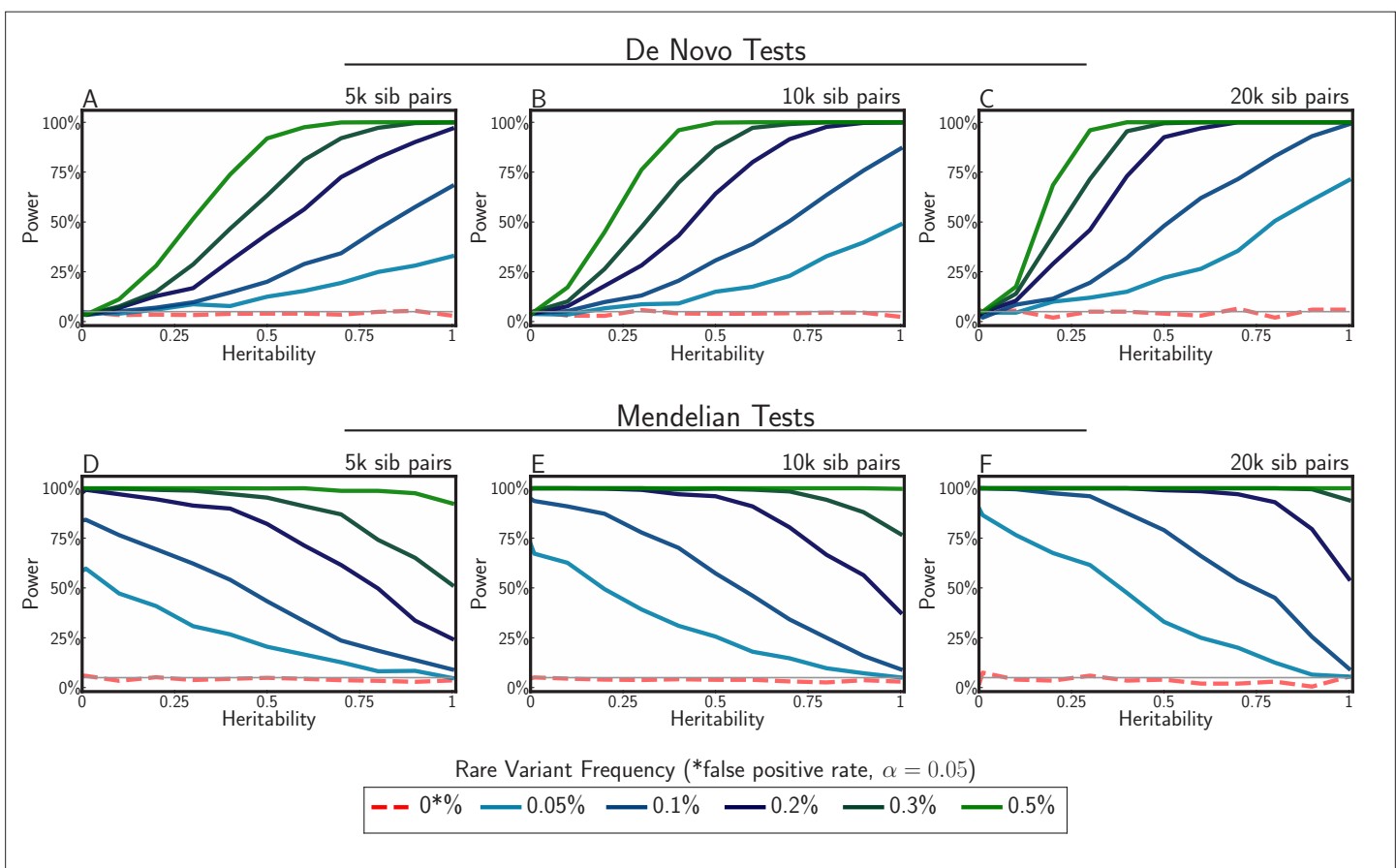

**Figure 5.** Power to detect complex tail architecture for different heritability levels, de novo and Mendelian frequencies, and sample sizes. Simulation assumes highly penetrant de novo and Mendelian frequencies of 0.05, 0.1, 0.2, 0.3, and 0.5%. The false-positive rate was set at 0.05. Null simulations (red dashed line) demonstrate tests are well calibrated.

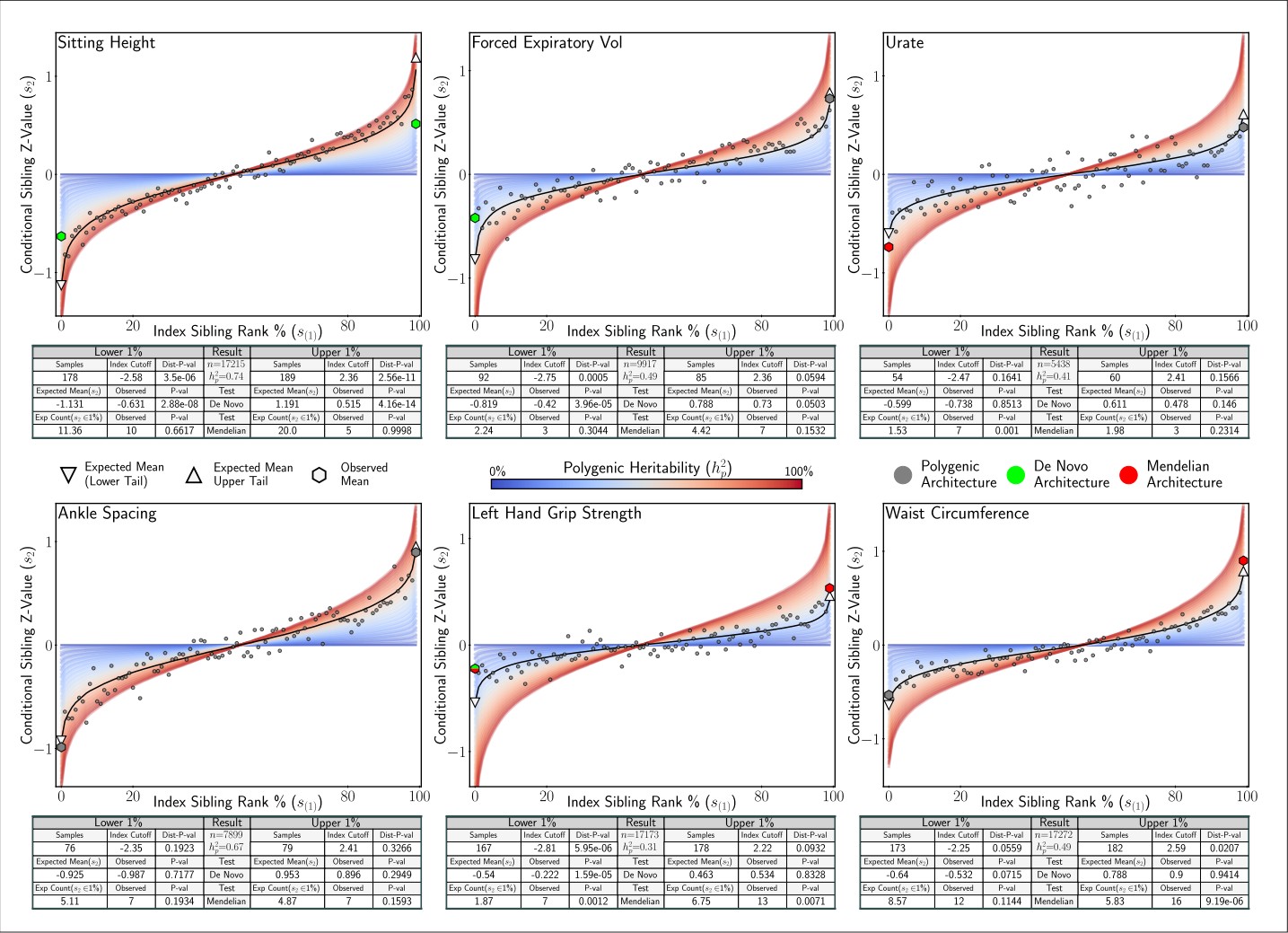

**Figure 6.** Analysis of six UK Biobank traits. Application of statistical tests for Mendelian and de novo tail architecture to sibling trait data of six UK Biobank traits. For each trait, the conditional sibling mean is plotted under polygenicity (black line) for the heritability estimated from the data. The red (high) and blue (low) bands represent the expected conditional sibling mean under polygenicity at different heritability values. Statistical tests for de novo architecture, Mendelian architecture, and general departure from polygenicity (Kolmogorov–Smirnov test, Dist P-val) were applied to conditional siblings with index siblings in the upper and lower 1% of the distribution. Significant associations for the Mendelian and de novo tests are shown in red and green, respectively. Tail architecture that is not distinct from polygenic expectation is denoted in grey.

In Appendix 2, we provide detailed derivations for the statistical tests described at a high level in the section 'Model and methods' and explain how they identify tail signatures in contrast to a polygenic background where conditional siblings regress-to-the-mean at a rate proportional to $h^2/2$ (*Equation 4*). The performance of the two tests evaluated using simulated sibling data is shown in *Figure 5*. These tests demonstrate that power to identify de novo architecture is greatest when heritability is high, while power to identify Mendelian architecture is greatest when heritability is low. These patterns can be explained by the fact that high heritability should lead to relatively high similarity among siblings, and low heritability to low similarity, under polygenicity. When heritability is estimated near 50% and at least 0.1% of the population has high-impact rare aetiology, both tests are well-powered to identify each class of complex tail architecture.

## Identifying complex tail architecture in UK Biobank data

We applied our statistical tests for complex tail architecture to sibling-pair data on 18 traits from the UK Biobank (*Sudlow et al., 2015*). Here (*Figure 6*), were present results from a set of six traits with varied tail architecture: *Sitting Height, Forced Expiratory Volume, Urate, Ankle Spacing, Left Hand*

*Grip Strength,* and *Waist Circumference.* For each trait, we estimated conditional sibling heritability via *Equation 5* and performed tests to identify de novo and Mendelian architecture in the lower and upper tails of the distribution of each trait. Additionally, we also performed a Kolmogorov–Smirnov test (*Marozzi, 2013*) to provide a general test for departures from our null model. We observed expected polygenic architecture in both tails for *Ankle Spacing.* We inferred Mendelian architecture in the lower tail for *Urate* and the upper tail for *Wait Circumference* and *Left Hand Grip Strength.* De novo architecture was inferred in the lower tail for *Forced Expiratory Volume* and strongly inferred in both tails for *Sitting Height,* which is supported by evidence from deep sequencing studies indicating that rare variants play a substantial role in the genetic aetiology for this trait (*Tcheandjieu et al., 2020*; *Bjornsdottir et al., 2022*). In the lower tail for *Left Hand Grip Strength,* we infer the presence of both de novo (greater than expected mean) and Mendelian (more concordant siblings than expected) architecture. We note that this could occur as a result of highly penetrant variants that are only shared among some siblings or perhaps because at the extremes siblings with different handedness are unexpectedly divergent and siblings with matching handedness are unexpectedly concordant. Extended results for all 18 traits analysed can be found in Appendix 4 (*Appendix 4—table 1*).

## Discussion

In this article, we present a novel approach to infer the genetic architecture of continuous traits, specifically in the tails of the distributions, from sibling trait data alone. Our approach is based on a theoretical framework that we develop, which derives the expected trait distributions of siblings conditional on the trait value of an index sibling and the trait heritability, assuming polygenicity, random mating, and environmental factors that are either controlled for or have combined Gaussian effects. The key intuition underlying the approach is that departures from the expected *conditional sibling trait distribution* in relation to index siblings selected from the trait tails may be due to non-polygenic architecture in the tails.

We demonstrate the validity of our conditional sibling analytical derivations through simulations and show that our tests for identifying de novo and Mendelian architecture in the tails of trait distributions are well-powered when large effect alleles are present in the population on the order of 1 out of 1000 individuals. Applying our test to a subset of well-powered traits in the UK Biobank, we find evidence of complex genetic architecture in at least one tail ($\alpha < 0.05$) in 16 of 18 traits and find de novo architecture occurring more frequently than Mendelian architecture (19 vs 6 of 36 total tails).

There are several areas in which our work could have short-term utility. Firstly, those individuals inferred as having rare variants of large effect could be followed up in multiple ways to gain individual-level insights. For example, they could undergo clinical genetic testing to identify potential pathogenic variants with effects beyond the examined trait, either in the form of diseases or disorders that the individual has already been diagnosed with or else that they have yet to present with but may be at high future risk for. Furthermore, investigation of their environmental risk profile may indicate an alternative – environmental – explanation for their extreme trait value (see below), rather than the rare genetic architecture inferred by our tests.

Our framework could also help refine the design of sequencing studies for identifying rare variants of large effect. Such studies either sequence entire cohorts at relatively high cost (*Uffelmann et al., 2021*) or else perform more targeted sequencing of individuals in the trait tails with the goal of optimising power per cost (*Yang et al., 2015*). However, even the latter approach is usually performed blind to evidence of enrichment of rare variant aetiology in the tails. Since our approach enables the identification of individuals that may be most likely to harbour rare variants, then these individuals could be prioritised for (deep) sequencing. Moreover, our ability to distinguish between de novo and Mendelian architecture could influence the broad study design, with the former suggesting that a family trio design may be more effective than population sequencing, which may be favoured if Mendelian architecture is inferred. Furthermore, our approach could be applied as a screening step to prioritise those traits, and corresponding tails, most likely to harbour rare variant architecture. Finally, if sequence data have already been collected, either cohort-wide or using a more targeted design, then our approach could be utilised to increase the power of statistical methods for detecting rare variants by upweighting individuals most likely to harbour rare variants.

This study has several limitations. First and foremost, departures from the expected conditional sibling trait distributions could be due to environmental risk factors, such as medication-use

or work-related exposures, rather than rare genetic architecture. Thus, while we believe that tail-specific deviations from polygenic expectation are interesting whether they arise primarily from genetic or environmental factors, we caution against over-interpretation of our results. Rejection of the null hypothesis from our tests should be considered only as indicating effects *consistent with* non-polygenic genetic architecture, alongside alternative explanations such as tail-specific unshared (de novo) or shared ('Mendelian') environmental risks. We suggest that further investigation of individuals' clinical, environmental, and genetic profiles is required to achieve greater certainty about the causes of their extreme trait values. Nevertheless, given knowledge that rare variants of large effect contribute to complex trait architecture, we expect that traits for which we infer non-polygenic architecture will, on average, be more enriched for rare architecture in the tail(s) than other traits. Secondly, our modelling assumes that the environmental risk factors of siblings are independent of each other. If in fact shared environmental risk factors contribute significantly to trait similarity among siblings, then our heritability estimates will be upwardly biased. However, this would only impact our tests if the degree of shared environmental risk differed in the tails relative to the rest of the trait distribution. Moreover, a large meta-analysis of heritability estimates from twin studies (*Polderman et al., 2015*) concluded that the contribution of shared environment among siblings (even twins) is insubstantial, and so we might expect this to have a limited impact on results from our tests. Thirdly, our modelling assumes random mating, and so the results from our tests in relation to traits that may be the subject of assortative mating should be considered with caution. Likewise, our modelling assumes additivity of genetic effects, and so, while additivity is well-supported by much statistical genetics research (*Wray et al., 2018*; *Hivert et al., 2021*), results from our tests should be reconsidered for any traits with evidence for significant non-additive genetic effects.

Our approach not only provides a novel way of inferring genetic architecture (without genetic data) but can do so specifically in the tails of trait distributions, which are most likely to harbour complex genetic architecture, due to selection, and are a key focus in biomedical research given their enrichment for disease. This work could also have broader implications in quantitative genetics since we derive fundamental results about the relationship among family members' complex trait values. The conditional sibling trait distribution provides a simple way of understanding the expected trait values of individuals according to their sibling's trait value, which could be used to answer questions of societal importance and inform future research. For example, it can be used to answer questions such as: as a consequence of genetics alone, how much overlap should there be in the traits of offspring of mid-parents at the 5th and 95th percentile and how does that contrast with what we observe in highly structured societies? Moreover, further development of the theory described here could lead to a range of other applications, for example, estimating levels of assortative mating, inferring historical selection pressures, and quantifying heritability in specific strata of the population.

## Acknowledgements

We thank the participants in the UK Biobank and the scientists involved in the construction of this resource for making the sibling data used in this article available. The work in this article has been conducted using the UK Biobank Resource under application 18177 (Dr O'Reilly). We would also like to thank Dr. Avi Reichenberg for early discussions and Dr. Peter Visscher for highlighting key references related to the topic, and Dr. Shai Carmi for providing feedback on a draft version of the article.

## Additional information

### Funding

No external funding was received for this work.

### Author contributions

Tade Souaiaia, Conceptualization, Software, Formal analysis, Validation, Investigation, Visualization, Methodology, Writing – original draft, Writing – review and editing; Hei Man Wu, Data curation, Formal analysis, Investigation, Project administration; Clive Hoggart, Conceptualization, Formal analysis, Methodology, Writing – original draft, Writing – review and editing; Paul F O'Reilly, Conceptualization,

Resources, Data curation, Supervision, Funding acquisition, Investigation, Writing – original draft, Project administration, Writing – review and editing

### Author ORCIDs
Tade Souaiaia (ORCID) https://orcid.org/0000-0003-3922-1372
Paul F O'Reilly (ORCID) https://orcid.org/0000-0001-7515-0845

Reviewer #1 (Public review): https://doi.org/10.7554/eLife.87522.3.sa1
Author response https://doi.org/10.7554/eLife.87522.3.sa2

## Additional files

### Supplementary files
MDAR checklist

### Data availability
Sibling data used is from UK Biobank (*Sudlow et al., 2015*) https://www.ukbiobank.ac.uk/ under Application 18177. The software produced, that can be used to carry out our analysis on sibling trait data, is made available at https://www.sibarc.net/ (https://github.com/tadesouaiaia/sibArc copy archived at *Souaiaia, 2025*).

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

## Appendix 1

### Derivation of conditional sibling distributions

Here, we derive the distribution that describes the probability density of the 'conditional' sibling ($S_2$) given the genetic liability, trait value, and case status of one or more index siblings. In each case, we assume a population of unrelated parents and rely on the results from the *infinitesimal polygenic model* that show that within family variance is normally distributed around mid-parent genetic liability (average of parents) with half the ancestral trait variance even when selection, drift, population structure, or dominance effects alter the between family trait distribution (**Barton and Keightley, 2002**; **Barton et al., 2017**; **Reichenberg et al., 2022**).

### Case (1) Index liability known, continuous trait ($h^2 = 1$): $P(S_2 \mid S_1 = s_1)$

We begin with the simplest case, a polygenic normally distributed trait which is fully heritable ($h^2 = 1$), where the genetic liability of an index sibling in a population is known. Throughout we denote the mid-parent, index sibling, and conditional sibling by $M, S_1$ and $S_2$. We begin by calculating the mid-parent distribution conditional on index liability using Bayes' theorem:

$$
\begin{aligned}
p(M = m \mid S = s) \quad &\propto p(S = s \mid M = m) \\
&= \mathcal{N}\left(S \mid m, \frac{\sigma^2}{2}\right)\mathcal{N}(M \mid 0, \sigma^2/2) \\
&\propto exp\left\{-\frac{(s-m)^2}{\sigma^2}\right\} exp\left\{-\frac{m^2}{\sigma^2}\right\} \\
&= \mathcal{N}(M \mid s/2, \sigma^2/4)
\end{aligned}
\tag{10}
$$

Then using the following identity (**Barndorff-Nielsen et al., 1982**)

$$
\int \mathcal{N}(x \mid \alpha z, \sigma_1^2)\mathcal{N}(z \mid \mu, \sigma_2^2)\, dz = \mathcal{N}(x \mid \alpha\mu, \sigma_1^2 + \alpha^2\sigma_2^2)
\tag{11}
$$

We calculate the conditional sibling distribution similarly:

$$
\begin{aligned}
p(S_2 \mid S_1 = s_1) \quad &= \int p(S_2, M = m \mid S_1 = s_1)\, dm \\
&= \int p(S_2 \mid M = m, S_1 = s_1)p(M = m \mid S_1 = s_1)\, dm \\
&= \int p(S_2 \mid M = m)p(M = m \mid S_1 = s_1)\, dm \\
&= \int \mathcal{N}(S_2 \mid m, \frac{\sigma^2}{2})\mathcal{N}(M = m \mid \frac{s_1}{2}, \frac{\sigma^2}{4})\, dm \\
p(S_2 \mid S_1 = s_1) \quad &= \mathcal{N}(S_2 \mid \frac{s_1}{2}, \frac{3\sigma^2}{4})
\end{aligned}
\tag{12}
$$

Thus, as predicted by the *infinitesimal polygenic model* the conditional sibling liability is normally distributed around the mid-parent liability distribution with additional variance equal to half the population liability variance.

### Case (2) Index trait value known, continuous trait ($h^2 \neq 1$): $P(S_2 \mid S_1 = s_1)$

In this case, the primary result considered in this article, a trait z-value, or equivalently, the percentile rank of an index sibling in genome-wide association where the rank-based inverse transformation has been applied (**McCaw et al., 2020**) is known. Transformation to a $Z$ distribution ($\sigma^2 = 1$) means that for heritability $h^2$ the genetic liability and environmental contributions to trait variance are $\sigma_g^2 = h^2$ and $\sigma_e^2 = 1 - h^2$, respectively. Similar to the previous case, we begin by calculating the conditional mid-parent liability from Bayes' theorem:

$$p(M_g = m_g \mid S_1 = s_1) \quad \propto p(S_1 = s_1 \mid M_g = m_g)$$

$$= \mathcal{N}(S_1 = s_1 \mid m_g, 1 - \frac{h^2}{2})\mathcal{N}(M_g = m_g \mid 0, h^2/2)$$

$$\propto exp\left\{-\frac{(s_1 - m)^2}{2 - h^2}\right\} exp\left\{-\frac{m_g^2}{h^2}\right\}$$

$$\propto exp\left\{-\frac{2m_g^2 - 2m_g s_1 h^2}{2h^2(1 - h^2/2)}\right\} \tag{13}$$

$$\propto exp\left\{-\frac{(m_g - s_1 h^2/2)^2}{h^2(1 - h^2/2)}\right\}$$

$$= \mathcal{N}(M_g = m_g \mid \frac{1}{2}s_1 h^2, \frac{1}{2}h^2(1 - h^2/2))$$

Then, we again use *Equation 11* to derive the distribution conditional sibling distribution:

$$p(S_2 = s_s \mid S_1 = s_1) = \int p(S_2 = s_2 \mid M_g = m_g)p(M_g = m_g \mid S_1 = s_1) \, dm_g$$

$$= \int \mathcal{N}\left(S_2 \mid m_g, 1 - \frac{h^2}{2}\right)\mathcal{N}\left(M_g \mid \frac{1}{2}s_1 h^2, \frac{1}{2}h^2\left(1 - \frac{h^2}{2}\right)\right) dm_g \tag{14}$$

$$= \mathcal{N}\left(S_2 = s_2 \mid \frac{1}{2}s_1 h^2, 1 - \frac{h^4}{4}\right)$$

## Case (3) Multiple index trait values known, continuous trait: $P(S_3 \mid S_1 = s_1, S_2 = s_2)$

The conditional sibling distribution can also be derived using the joint trait distribution for related individuals:

$$y \sim \mathcal{N}(0, \mathbf{G})$$

where the covariance $\mathbf{G}$ is the genetic relationship matrix. Thus for a sibling pair

$$p(s_1, s_2) = \mathcal{N}_2\left(0, \begin{pmatrix} 1 & h^2/2 \\ h^2/2 & 1 \end{pmatrix}\right)$$

As shown by *Bernardo and Smith, 1994*, if $X$ is multivariate normal $\mathcal{N}(\mu, \lambda^{-1})$, where $\lambda = \Sigma^{-1}$ is the precision matrix, and $X$ is partitioned into $x_1$ and $x_2$, with corresponding partitions of μ and λ of

$$\mu = \begin{pmatrix} \mu_1 \\ \mu_2 \end{pmatrix}, \quad \lambda = \begin{pmatrix} \lambda_{11} & \lambda_{12} \\ \lambda_{21} & \lambda_{22} \end{pmatrix}$$

then the conditional distribution of $x_1$ given $x_2$ is also normal with mean and precision matrix:

$$\mu_1 - \lambda_{11}^{-1}\lambda_{12}(x_2 - \mu_2), \quad \lambda_{11} \tag{15}$$

Thus, given the joint distribution for three siblings:

$$p(s_1, s_2, s_3) = \mathcal{N}_3\left(0, \begin{pmatrix} 1 & h^2/2 & h^2/2 \\ h^2/2 & 1 & h^2/2 \\ h^2/2 & h^2/2 & 1 \end{pmatrix}\right)$$

The precision matrix $\lambda = \Sigma^{-1}$ is

$$\lambda = \frac{1}{2 + h^2 - h^4} \begin{bmatrix} 2+h^2 & -h^2 & -h^2 \\ -h^2 & 2+h^2 & -h^2 \\ -h^2 & -h^2 & 2+h^2 \end{bmatrix}$$

And we can calculate the conditional distribution for two sibling using *Equation 15*:

$$p(S_3 \mid S_1 = s_1, S_2 = s_2) = \mathcal{N}\left(S_3 \mid \frac{h^2}{2 + h^2}(s_1 + s_2), 1 - \frac{h^4}{2 + h^2}\right)$$

## Case (4) Binary trait: $P(S_2 \mid S_1 = \textit{Affected})$

Here, we again assume an underlying distribution that is $\mathcal{N}(0, 1)$ and made up of genetic and environmental components. However, we only know the index sibling's *status*, which, as described under the liability threshold model (**Falconer, 1967**), is equivalent to conditioning on the event where than index sibling's trait value is above or below a z-value threshold $T$:

$$p(S_2 \mid S_1 = \text{Affected}) = p(S_2 \mid S_1 > T) \text{ and } p(S_2 \mid S_1 = \cancel{\text{Affected}}) = p(S_2 \mid S_1 < T) \tag{16}$$

where $T = \Phi^{-1}(1 - K)$, $\Phi^{-1}$ is the inverse normal cumulative distribution function, and $K$ is the incidence of the binary trait in the population. Thus, the conditional distribution one sibling given an *Affected* index sibling can be calculated integrating over the normal distribution truncated at $T$:

$$p(S_2 \mid S_1 > T) = \int_T^\infty p(S_2 \mid S_1 = s_1)p(S_1 = s_1)ds_1 \tag{17}$$

The first two moments of this truncated normal (**Johnson et al., 1995**) are

$$E(S_1 \mid S_1 > T) = \frac{\phi(T)}{1 - \Phi(T)} = \frac{\phi(T)}{K} = i$$

$$V(S_1 \mid S_1 > T) = 1 + T\frac{\phi(T)}{K} - \left(\frac{\phi(T)}{K}\right)^2 = 1 + iT - i^2$$

Approximating the index sibling distribution using a normal whose moments are taken from this truncated distribution, **Equation 17** becomes

$$p(S_2 = s_s \mid S_1 > T) = \int \mathcal{N}\left(S_2 = s_2 \mid \frac{1}{2}s_{1h}^{*}{}^2, 1 - \frac{h^4}{4}\right) \mathcal{N}(S_1 = s_1^* \mid i, 1 + iT - i^2)ds^{*1} \tag{18}$$

which can be solved using the identity given in **Equation 11**:

$$\begin{aligned} &= \mathcal{N}\left(S_2 = s_2 \mid \frac{1}{2}ihx^2, 1 - \frac{h^4}{4} + \frac{1}{4}h^4(1 + iT - i^2)\right) \\ &= \mathcal{N}\left(S_2 = s_2 \mid \frac{1}{2}ih^2, 1 - \frac{h^4}{4}i(i - T)\right) \end{aligned} \tag{19}$$

Thus, conditional on an *Affected* sibling, the probability of concordance is

$$p(S_2 = \text{Affected} \mid S_1 = \text{Affected}) = p(S_2 > T \mid S_1 > T)$$

$$= 1 - \Phi\left(\frac{T - ih^2/2}{\sqrt{1 - h^4 i(i - T)/4}}\right)$$

which is equivalent to Reich's (**Reich et al., 1972**) correction to Falconer's (**Falconer, 1965**) approximation where the relationship between the relatives is 0.5 for siblings. The probability of discordance given an *Unaffected* sibling can be calculated from Bayes' theorem:

$$p(S_2 = \text{Affected} \mid S_1 = \cancel{\text{Affected}}) = \frac{p(S_1 = \cancel{\text{Affected}} \mid S_2 = \text{Affected})\,p(S_2 = \text{Affected})}{p(S_1 = \cancel{\text{Affected}})}$$

$$= \frac{K - Kp(S_2 = \text{Affected} \mid S_1 = \text{Affected})}{1 - K}$$

which allow the conditional probability of case status to be determined given an index sibling's status.

## Appendix 2

## Statistical tests for complex architecture

Here, we describe our statistical tests for complex tail architecture. Our tests identify changes in the conditional sibling distribution when ascertaining an index sibling in the tail relative to polygenic expectation. To carry out these tests, we establish a null distribution built on the assumption that indexing on siblings not in the tails reduces the likelihood that either sibling phenotype in the pair is driven by rare variants of large effect. We use the region from the 5th to the 95th percentile to estimate heritability. From the $n$ sibling pairs where the index sibling $s_1^{(i)}$ is in the 5th to the 95th, we calculate the conditional likelihood *Equation 14*:

$$\mathcal{L}(D|h^2) = \prod_i^n p(s_2^{(i)}|s_1^{(i)}, h^2) \propto \left(1 - \frac{h^4}{4}\right)^{-n/2} \exp\left(-\frac{\sum_i^n \left(s_2^{(i)} - \frac{s_1^{(i)}h^2}{2}\right)^2}{2\left(1 - \frac{h^4}{4}\right)}\right) \tag{20}$$

and maximise the log-likelihood with respect to $h^2$:

$$\log \mathcal{L} \propto -n \log\left(1 - \frac{h^4}{4}\right) - \frac{1}{1 - \frac{h^4}{4}} \sum_{i=1}^{n} \left(s_2^{(i)} - \frac{s_1^{(i)}h^2}{2}\right)^2 \tag{21}$$

to obtain a maximum likelihood estimate for $h^2$ that is used to define the null distribution for our statistical tests.

### Statistical test for de novo architecture

We identify de novo mutations of large effect by testing for discordance between siblings relative to the polygenic null using the conditional distribution of a sibling given index sibling from *Equation 14*. Since de novo mutations typically result in trait values in the tail of the distribution, the test conditions on those index siblings in a specified upper quantile $q$ of the distribution, that is, those sibling pairs $(s_1^{(i)}, s_2^{(i)})$ such that $s_1^{(i)} > \Phi(q)$, defined as the set $A_q$. We introduce an additional parameter $\alpha$ where values of $\alpha < 0$ in the right tail and $\alpha > 0$ in the left tail are indicative of discordant siblings with trait values closer to the mean, giving a log-likelihood:

$$\log \mathcal{L} = -\frac{1}{2(1 - \frac{h^4}{4})} \sum_{i \in A_q} \left(s_2^{(i)} - \left(\frac{1}{2}s_1^{(i)}h^2 + \alpha\right)\right)^2 \tag{22}$$

The null hypothesis $H_0 : \alpha = 0$ is tested via a score test:

$$U = \frac{d}{d\alpha} log \mathcal{L} = \frac{1}{1 - \frac{h^4}{4}} \sum_{i \in A} s_2^{(i)} - \left(\frac{1}{2}s_2^{(i)}h^2 + \alpha\right) \tag{23}$$

$$I = \frac{d^2}{d\alpha^2} \log \mathcal{L} = -\frac{n}{1 - \frac{h^4}{4}} \tag{24}$$

And the score test for $H_0 : \alpha = 0$:

$$z = \frac{U}{\sqrt{-I}} = \frac{\sum_{i \in A_q} s_2^{(i)} - \frac{1}{2} s_2^{(i)} h^2}{\sqrt{n(1 - \frac{h^4}{4})}} \tag{25}$$

## Statistical test for mendelian architecture

Here, we test for excess concordance between siblings in the tails of the distribution by testing for an excess number of observed siblings in the tail $S_2 > \Phi^{-1}(q)$ given the index sibling is in the tail $S_1 > \Phi^{-1}(q)$, where $q$ is the quantile of interest. Denoting the set of index siblings in the tail by $A_q$ and the size of the set by $n$, under the null of pologenicity, we calculate the probability that the conditional sibling exceeds $\Phi^{-1}(q)$ from the normal cdf and compute the mean to define mean concordance under polygenicity $\pi_0$:

$$\pi_0 = \frac{1}{n} \sum_{i \in A_q} P \left( S_2^{(i)} > \Phi^{-1}(q) \mid S_1^{(i)} = s_1^{(i)} \right)$$

$$= \frac{1}{n} \sum_{i \in A_q} \left( 1 - \Phi \left( \frac{\Phi^{-1}(q) - \frac{s_1^{(i)} h^2}{2}}{\sqrt{1 - \frac{h^4}{4}}} \right) \right)$$

Denoting the observed concordance (number of sibling pairs both $> \Phi^{-1}(q)$) by $r$, the binomial log-likelihood (ignoring the constant) is

$$\log \mathcal{L} = r \log \pi + (n - r) \log(1 - \pi)$$

$$U = \frac{d}{d\pi} \log L = \frac{r}{\pi} - \frac{n - r}{1 - \pi}$$

$$= \frac{r(1 - \pi) - (n - r)\pi}{\pi(1 - \pi)}$$

$$= \frac{r - n\pi}{\pi(1 - \pi)}$$

$$I = \frac{d^2}{d\pi^2} \log L = -\frac{r}{\pi^2} - \frac{n - r}{(1 - \pi)^2}$$

$$= -\frac{r(1 - \pi)^2 + (n - r)\pi^2}{\pi^2(1 - \pi)^2}$$

Assuming $r = n\pi$ such that $I$ is not a function of any particular observation:

$$= -\frac{n}{\pi(1 - \pi)}$$

And the score test for $H_0 : \pi = \pi_0$:

$$z = \frac{U}{\sqrt{-I}} = \frac{r - n\pi_0}{\sqrt{n\pi_0(1 - \pi_0)}}$$

## Statistical test for non-polygenic architecture

General departures from polygenicity can be identified based on the degree of discordance in conditional sibling distribution from the expectation. Assuming polygenicity, given index and conditional sibling pairs $(s_1^{(i)}, s_2^{(i)})$, from **Equation 4**, we can write

$$z_2^{(i)} = \frac{(s_2^{(i)} - \frac{1}{2} s_1^{(i)} h^2)}{\sqrt{1 - \frac{h^4}{4}}} \sim \mathcal{N}(0, 1)$$

Therefore, departures in polygenicity can be tested in trait quantiles by testing the observed distribution of $z_2^{(i)}, i \in A_q$, where $A_q$ is the set of index siblings $s_1^{(i)}$ in the quantile of interest, relative to a standard normal distribution via the Kolmogorov–Smirnov test.

# Appendix 3

## Model evaluation

Here, we compare our theoretical derivations (*Equation 4*) that rely on the *infinitesimal polygenic model* (*Barton et al., 2017*) with simulated offspring data (*Appendix 3—figure 1*). We also compare our model to our empirical simulation (see 'Model and methods') that draws allele frequencies and effect sizes from publicly available GWAS data (*Neale, 2018*) for two traits to produce parent and offspring genotype and genetic liability (equivalent to trait value when $h^2 = 1$, *Appendix 3—figure 2*).

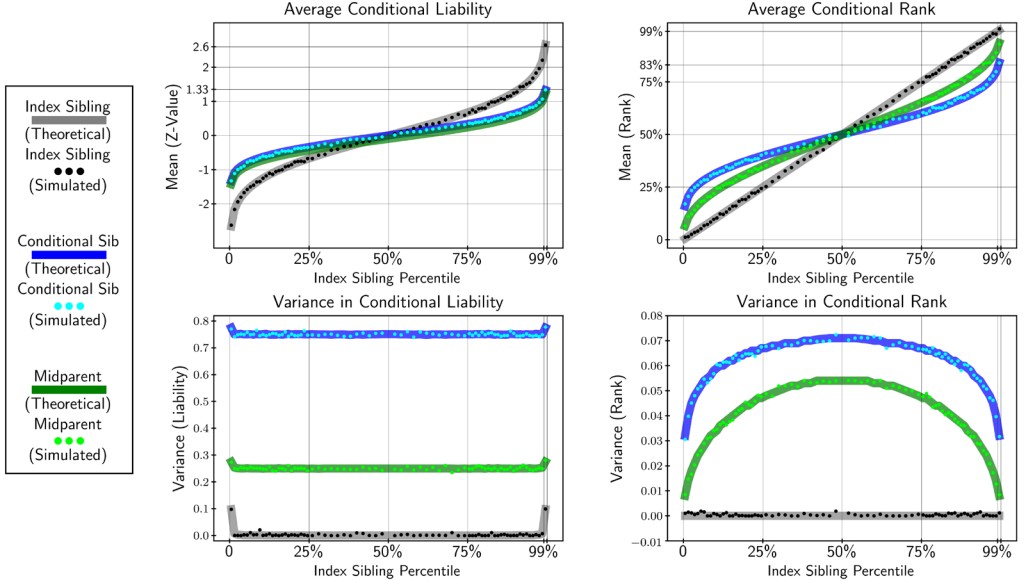

**Appendix 3—figure 1.** Theoretical and simulated conditional expectation and variance in liability (z-score) and rank across index sibling percentiles for conditional sibling, mid-parents, and index siblings. Simulation drew 1000 parent liability values from, $\mathcal{N}(0, 1)$; these were randomly paired to produce to mid-parents with liability, $m_i$; two offspring were subsequently drawn from $\left(\mathcal{N}(m_i, \frac{1}{2})\right)$ and randomly assigned as index and conditional siblings.

*Appendix 3—figure 2 continued on next page*

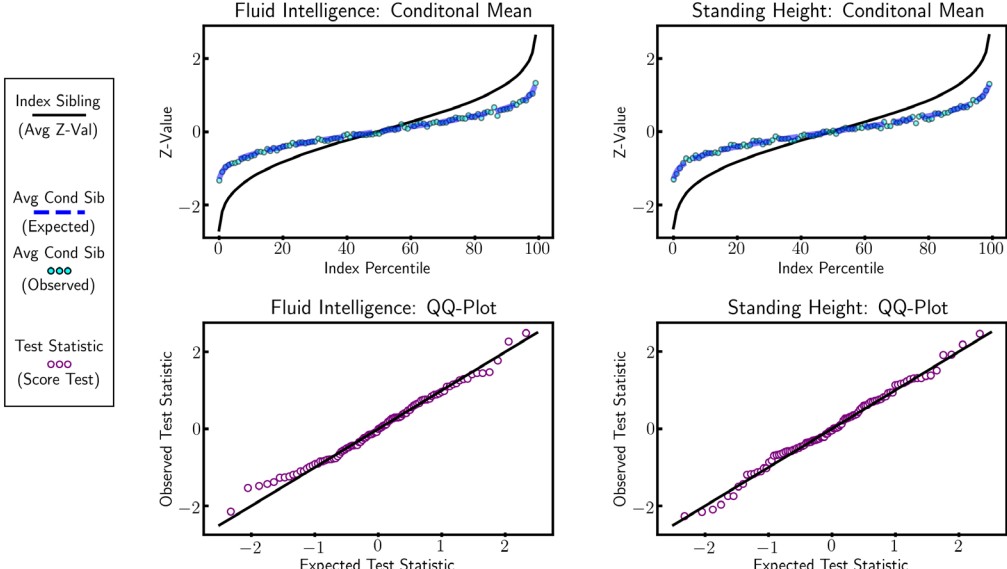

**Appendix 3—figure 2.** For both *Fluid Intelligence* and *Standing Height,* genome-wide association studies (GWAS) variants (on chromosome 1) were used to simulate parent and offspring genotypes and liability values. Plots show that for both traits the offspring distribution is normal and that the sibling distribution is multivariate normal, in line with our theoretical prediction.

These tests demonstrate that our theoretical framework accurately reflects an additive polygenic trait in an outcrossing population. Additionally, these results demonstrate that deviations in the conditional sibling distribution can be interpreted as non-polygenic architecture or quantile-specific environmental effects.

# Appendix 4

## Software availability and extended results

We have made our code, sample data, and a brief tutorial available online at https://www.sibarc.net/ (*sibArc:Software for Inference of Genetic Architecture, 2024*). Below, we display the trait summaries and sibling test results for all 18 UK Biobank traits analysed and referenced in the article.

**Appendix 4—table 1.** Application to the UK Biobank (extended result – trait summaries).

| Trait name (field ID) | Sib pairs | Unique values | Skew | Kurtosis | Sib-$h^2$ (full) | Sib-$h^2$ (5–95) |
|---|---|---|---|---|---|---|
| LH Grip Strength (46) | 17,174 | 85 | 0.369 | 2.731 | 0.32 | 0.36 |
| Waist Circumference (48) | 17,273 | 716 | 0.42 | 3.29 | 0.49 | 0.52 |
| Ankle Spacing (3143) | 7900 | 1928 | 0.25 | 3.11 | 0.67 | 0.69 |
| Sitting Height (20015) | 17,216 | 352 | 0.04 | 3.4 | 0.73 | 0.82 |
| Forced Expiratory Vol (20150) | 9918 | 662 | 0.4 | 3.22 | 0.49 | 0.58 |
| Body Fat % (23099) | 16,785 | 577 | 0.09 | 2.56 | 0.53 | 0.57 |
| Whole Body Impedance (23106) | 16,801 | 771 | 0.24 | 2.8 | 0.57 | 0.59 |
| Right Leg Impedance (23107) | 16,793 | 350 | 0.15 | 3.34 | 0.55 | 0.58 |
| Left Leg Impedance (23108) | 16,793 | 352 | 0.14 | 3.37 | 0.54 | 0.56 |
| Right Arm Impedance (23109) | 16,983 | 475 | 0.33 | 2.63 | 0.51 | 0.53 |
| Left Arm Impedance (23110) | 16,797 | 455 | 0.34 | 2.62 | 0.51 | 0.53 |
| Trunk Fat Percentage (23127) | 16,780 | 644 | -0.08 | 3.19 | 0.51 | 0.52 |
| Red Blood Cell Count (30010) | 16,307 | 3421 | 0.11 | 3.31 | 0.48 | 0.53 |
| Haemoglobin Concentration (30020) | 16,307 | 1119 | -0.07 | 3.36 | 0.39 | 0.44 |
| Haematocrit Percentage (30030) | 16,303 | 2872 | -0.02 | 3.35 | 0.37 | 0.41 |
| Cholesterol (30690) | 5452 | 7028 | 0.37 | 3.44 | 0.47 | 0.53 |
| LDL Direct (30780) | 5432 | 5475 | 0.37 | 3.36 | 0.45 | 0.5 |
| Urate (30880) | 5439 | 5035 | 0.46 | 3.16 | 0.42 | 0.42 |

**Appendix 4—table 2.** Application to the UK Biobank (extended result – sibling tests).

| Trait name (field ID) | Tail | Idx sib cutoff | KS test p value | De novo obs, exp | De novo p value | *Mendelian* obs, exp | *Mend* p value |
|---|---|---|---|---|---|---|---|
| LH Grip Strength (46) | Upper | 2.22 | 0.0932 | 0.53, 0.46 | 0.8328 | 13, 6.8 | 0.0071 |
| | Lower | −2.81 | 5.95e-06 | −0.22, −0.54 | 1.59e-05 | 7, 1.9 | 0.0012 |
| Waist Circ. (48) | Upper | 2.59 | 0.0207 | 0.9, 0.79 | 0.9414 | 16, 5.8 | 9.1e-06 |
| | Lower | −2.25 | 0.0559 | −0.53, −0.64 | 0.0715 | 12, 8.6 | 0.1144 |
| Ankle Spacing (3143) | Upper | 2.41 | 0.3266 | 0.9, 0.95 | 0.2949 | 7, 4.9 | 0.1593 |
| | Lower | −2.35 | 0.1923 | −0.99, −0.93 | 0.7177 | 7, 5.1 | 0.1934 |
| Sitting Height (20015) | Upper | 2.36 | 2.56e-11 | 0.52, 1.19 | 4.16e-14 | 5, 20.0 | 0.9998 |
| | Lower | −2.58 | 3.5e-06 | −0.63, −1.13 | 2.88e-08 | 10, 11.4 | 0.6617 |
| FEV (20150) | Upper | 2.36 | 0.0594 | 0.73, 0.79 | 0.0503 | 7, 4.4 | 0.1532 |
| | Lower | -2.75 | 0.0005 | −0.42, −0.82 | 3.96e-05 | 3, 2.2 | 0.3044 |
| Body Fat (23099) | Upper | 2.33 | 0.5765 | 0.88, 0.79 | 0.8754 | 17, 9.2 | 0.0044 |
| | Lower | −2.55 | 0.004 | −0.65, −0.85 | 0.0032 | 9, 6.6 | 0.1748 |
| Body Imp. (23106) | Upper | 2.4 | 0.0565 | 0.68, 0.83 | 0.0198 | 7, 8.9 | 0.7465 |
| | Lower | −2.46 | 0.012 | −0.71, −0.87 | 0.0209 | 10, 8.2 | 0.2566 |

*Appendix 4—table 2 Continued on next page*

*Appendix 4—table 2 Continued*

| Trait name (field ID) | Tail | Idx sib cutoff | KS test p value | De novo obs, exp | De novo p value | *Mendelian* obs, exp | *Mend* p value |
|---|---|---|---|---|---|---|---|
| Right Leg Imp. (23107) | Upper | 2.31 | 0.7491 | 0.76, 0.78 | 0.3919 | 9, 9.9 | 0.6143 |
| | Lower | −2.5 | 0.0003 | −0.59, −0.85 | 0.0003 | 6, 7.2 | 0.672 |
| Left Leg Imp. (23108) | Upper | 2.3 | 0.1393 | 0.83, 0.74 | 0.8965 | 10, 9.1 | 0.3752 |
| | Lower | −2.5 | 0.0093 | −0.65, −0.81 | 0.0181 | 5, 6.3 | 0.7078 |
| Right Arm Imp. (23109) | Upper | 2.51 | 0.001 | 0.58, 0.81 | 0.0012 | 3, 6.9 | 0.9356 |
| | Lower | −2.37 | 0.4972 | −0.72, −0.77 | 0.252 | 10, 8.7 | 0.3251 |
| Left Arm Imp (23110) | Upper | 2.54 | 0.0023 | 0.66, 0.81 | 0.0306 | 7, 6.1 | 0.3548 |
| | Lower | −2.4 | 0.599 | −0.69, −0.76 | 0.1767 | 10, 7.9 | 0.2183 |
| Trunk Fat % (23127) | Upper | 2.21 | 0.5881 | 0.68, 0.66 | 0.6269 | 11, 8.8 | 0.2204 |
| | Lower | −2.51 | 0.1247 | −0.65, −0.74 | 0.1056 | 9, 5.5 | 0.0624 |
| RBC Count (30010) | Upper | 2.29 | 0.0892 | 0.57, 0.72 | 0.0256 | 8, 9.2 | 0.653 |
| | Lower | −2.49 | 1.58e-09 | −0.34, -0.82 | 1.07e-10 | 4, 7.1 | 0.8815 |
| Haemoglobin Conc (30020) | Upper | 2.28 | 0.0217 | 0.49, 0.6 | 0.0643 | 4, 7.2 | 0.8908 |
| | Lower | −2.59 | 4.34e-06 | −0.32, −0.71 | 1.67e-07 | 5, 4.4 | 0.3878 |
| Haematocrit % (30030) | Upper | 2.29 | 0.0104 | 0.34, 0.55 | 0.0034 | 3, 6.4 | 0.9135 |
| | Lower | −2.52 | 0.0004 | −0.35, −0.63 | 0.0002 | 4, 4.3 | 0.5622 |
| Cholesterol (30690) | Upper | 2.28 | 0.4059 | 0.57, 0.71 | 0.1425 | 6, 3.1 | 0.0479 |
| | Lower | −2.35 | 0.0045 | −0.37, −0.74 | 0.0012 | 5, 3.1 | 0.1316 |
| LDL Direct (30780) | Upper | 2.3 | 0.2262 | 0.37, 0.66 | 0.0083 | 3, 3.0 | 0.5098 |
| | Lower | −2.35 | 0.0112 | −0.51, −0.68 | 0.0918 | 5, 2.6 | 0.0614 |
| Urate (30880) | Upper | 2.41 | 0.1566 | 0.48, 0.61 | 0.146 | 3, 2.0 | 0.2314 |
| | Lower | −2.47 | 0.1641 | −0.74, -0.6 | 0.8513 | 7, 1.5 | 0.001 |

