## [Editor Report · eLife assessment]

The authors present a **solid** statistical framework for using sibling phenotype data to assess whether there is evidence for de novo or rare variants causing extreme trait values. Their **valuable** method is promising and will be of interest to researchers studying complex trait genetics.

---

## [Referee Report · Reviewer #1 (Public review)]

This is a clever and well-done paper. The authors sought to craft a method, applicable to biobank-scale data but without necessarily using genotyping or sequencing, to detect the presence of de novo mutations and rare variants that stand out from the polygenic background of a given trait. Their method depends essentially on sibling pairs where one sibling is in an extreme tail of the phenotypic distribution and whether the other sibling's regression to the mean shows a systematic deviation from what is expected under a simple polygenic architecture.

Their method is successful in that it builds on a compelling intuition, rests on a rigorous derivation, and seems to show reasonable statistical power in the UK Biobank. (More biobanks of this size will probably become available in the near future.) It is somewhat unsuccessful in that rejection of the null hypothesis does not necessarily point to the favored hypothesis of de novo or rare variants. The authors discuss the alternative possibility of rare environmental events of large effect.

Comments on current version:

The authors have addressed the concerns of the reviewers. I have no further comments.

---

## [Author Response]

The following is the authors’ response to the original reviews.

**eLife assessment**
The authors present valuable findings on how to determine the genetic architecture of extreme phenotype values by using data on sibling pairs. While the authors' derivations of the method are correct, the scenarios considered are incomplete, making it difficult to have confidence in the interpretation of the results as demonstrating the influence of de-novo or Mendelian (rare, penetrant-variant) architectures. The method nevertheless shows promise and will be of interest to researchers studying complex trait genetics.

A.1: We have now expanded our consideration of the scenarios and we have ensured that we do not over-interpret our results as being due to de novo or Mendelian architectures. Instead, we make clear that our statistical tests are powered to identify these architectures but that there are other potential causes of significant results (e.g. measurement error or uncontrolled environmental factors from heavy-tailed distributions), making follow-up validation studies necessary before underlying architectures can be confirmed. We consider this to be typical of observational research, in which significant results may indicate causal effects unless uncontrolled confounding factors explain the observed associations, requiring experimental/trial follow-up for validation. We believe that our tests are useful for providing initial inference, and that in some settings – e.g. prioritising samples for sequencing to identify rare variants – could be useful as an initial screening step to increase the efficacy of a planned analysis or study.

Additionally, we have now developed “SibArc”, an openly available software tool that takes input sibling trait data and estimates conditional sibling heritability across the trait distribution. Then - based on our theoretical framework developed and described in the paper - for each tail of the trait distribution, estimates effect sizes and generates P-values corresponding to our de novo and Mendelian tests, and performs a Kolmogorov-Smirnov test to identify general departures from our null model. Furthermore, SibArc also provides additional functionality for users under preliminary beta form, for example, running an iterative optimisation routine to infer approximate relative degrees of polygenic, de novo, and Mendelian architectures prevailing in each trait tail. We have made this software tool, Quick Start tutorial, and sample data available online at Github and are hosting these on a dedicated website: www.sibarc.net.

**Reviewer #1 (Public Review):**
This is a clever and well-done paper that should be published. The authors sought to craft a method, applicable to biobank-scale data but without necessarily using genotyping or sequencing, to detect the presence of de novo mutations and rare variants that stand out from the polygenic background of a given trait. Their method depends essentially on sibling pairs where one sibling is in an extreme tail of the phenotypic distribution and whether the other sibling's regression to the mean shows a systematic deviation from what is expected under a simple polygenic architecture.Their method is successful in that it builds on a compelling intuition, rests on a rigorous derivation, and seems to show reasonable statistical power in the UK Biobank. (More biobanks of this size will probably become available in the near future.) It is somewhat unsuccessful in that rejection of the null hypothesis does not necessarily point to the favored hypothesis of de novo or rare variants. The authors discuss the alternative possibility of rare environmental events of large effect. Maybe attention should be drawn to this in the abstract or the introduction of the paper. Nevertheless, since either of these possibilities is interesting, the method remains valuable.

A.2: We agree with the reviewer that we should have made it clearer that - while our statistical tests are powered to identify de novo and Mendelian architectures – significant findings from our tests could also be explained by rare environmental events of large effect (specifically by uncontrolled environmental factors with heavy-tailed distributions). We have now made this clear throughout the manuscript (see A.1).

Moreover, we agree with the reviewer that whether the cause of deviations from expectations are due to de novo or rare variants, or environmental factors, either possibility is interesting. For example, in either scenario, our results can highlight inaccuracy in PRS prediction of extreme trait values for certain traits, and also provides a relative measure across different traits of large effects impacting on the trait tails, irrespective of whether genetic or environmental. We now place more emphasis on this point throughout the manuscript.

**Reviewer #2 (Public Review):**
Souaiaia et al. attempt to use sibling phenotype data to infer aspects of genetic architecture affecting the extremes of the trait distribution. They do this by considering deviations from the expected joint distribution of siblings' phenotypes under the standard additive genetic model, which forms their null model. They ascribe excess similarity compared to the null as due to rare variants shared between siblings (which they term 'Mendelian') and excess dissimilarity as due to de-novo variants. While this is a nice idea, there can be many explanations for rejection of their null model, which clouds interpretation of Souaiaia et al.'s empirical results.

A.3: We agree with the reviewer that we should have made clearer that there are other explanations for significant results from our tests and we have now fully addressed this point – (see A.1, A.2, A.4, A.5 for more detail). In addition, we now elaborate on exactly what our null hypothesis is: which is not only that the expected joint distribution of siblings’ phenotypes is governed by the standard additive genetic model, but that environmental effects are either controlled for or else their combined effect is approximately Gaussian. Furthermore, by selecting only those traits whose raw trait distribution most closely corresponds to a Gaussian distribution from the UK Biobank, we increase the probability that significant results from our tests are due to rare variants (shared or unshared among siblings).

The authors present their method as detecting aspects of genetic architecture affecting the extremes of the trait distribution. However, I think it would be better to characterize the method as detecting whether siblings are more or less likely to be aggregated in the extremes of the phenotype distribution than would be predicted under a common variant, additive genetic model.

A.4: As discussed above we should have stated more clearly that significant results could be due to non-genetic factors, we have now addressed this.

However, we do not think that it would be appropriate to characterise our tests as merely corresponding to over and under aggregation of siblings in the tails. Firstly, environmental factors should be controlled for as part of our testing, increasing the probability that significant results are due to genetic, and not environmental factors. Secondly, tests for identifying broad over and under aggregation of siblings in the tails should be designed differently and, accordingly, the tests that we have developed here would not be optimal to detect over/under aggregation of siblings in trait tails. Our test for inference of de novo variants, for example, exploits the fact that de novo alleles of large effect result in one sibling being extreme and all others being drawn from the background distribution, so that the mean of other siblings is relatively low – not merely that other siblings are less likely to be found in the tail. For more discussion on this issue in relation to one of reviewer 1’s points, see A.9.

Exactly how the rareness and penetrance of a genetic variant influence the conditional sibling phenotype distribution at the extremes is not made clear. The contrast between de-novo and 'Mendelian' architectures is somewhat odd since these are highly related phenomena: a 'Mendelian' architecture could be due to a de-novo variant of the previous generation. The fact that these two phenomena are surmised to give opposing signatures in the authors' statistical tests seems suboptimal to me: would it not be better to specify a parameter that characterizes the degree or sharing between siblings of rare factors of large effect? This could be related to the mixture components in the bimodal distribution displayed in Fig 1. In fact, won't the extremes of all phenotypes be influenced by all three types of variants (common, rare, de-novo) to greater or lesser degree? By framing the problem as a hypothesis testing problem, I think the authors are obscuring the fact that the extremes of real phenotypes likely reflect a mixture of causes: common, de-novo, and rare variants (and shared and non-shared environmental factors).

A.5: We absolutely recognise that there will typically be a complex and continuous mix of genetic architectures underlying complex traits in their tails, dictated by the 2-dimensional relationship between allele frequency and effect size. We did consider developing a fully Bayesian statistical framework to model this, but soon realised that doing this properly would require a substantial amount of model development, accounting for multiple factors in ways that would require a great deal of further investigation; for example, performing a range of complex simulations to investigate the effects of different selective pressures over time, different patterns of assortative mating, and effect size generating distributions. We are in the process of applying for funding for a multi-year project that will perform exactly these investigations as a step towards developing more sophisticated models of inference. In the meantime, we do believe that the simpler hypothesis-testing framework that we have developed here does have important value. Assuming that environmental factors are accounted for, or that any that are not accounted for have combined Gaussian effects, then our tests will indeed infer enrichments of de novo and ‘Mendelian’ rare alleles of large effect in the tails of complex traits. Results from these tests can also be compared within and across traits to compare the relative degree of such enrichments among traits. For some traits we observe significant results from both tests, and for other traits we observe highly significant results from one of our tests but not the other. Thus, while our tests do not provide a complete picture about the genetic architecture in the tails of complex traits, they do offer some intriguing initial insights into tail architecture, important given the enrichment of disease in trait tails.

To better enable interpretation of the results of this method, a more comprehensive set of simulations is needed. Factors that may influence the conditional distribution of siblings' phenotypes beyond those considered include: non-normal distribution, assortative mating, shared environment, interactions between genetic and shared environmental factors, and genetic interactions.

A.6: As described above (see A.5) we do agree that a more comprehensive set of simulations is exactly what is needed to further extend this work. However, we believe that the tests that we have developed so far, which make some simplifying assumptions that we think would often hold in practice, is a useful start to what is an entirely novel approach to inferring genetic architecture from family trait-only (non-genetic) data. Our work could already be useful for method developers who may wish to extend our approach in ways that we may not think of. It could also be useful for applied scientists focusing on specific traits who will be able to gain initial, inference-level, insights by applying our tests to their data, while the results of applying our tests may even guide study design of rare variant mapping studies.

In summary, I think this is a promising method that is revealing something interesting about extreme values of phenotypes. Determining exactly what is being revealed is going to take a lot more work, however.

A.7: We thank the reviewer for highlighting the promise in our approach and agree that it is revealing something interesting about complex traits. We also agree that it is going to take a lot more work to reveal exactly what that is for different traits, which we plan to work on ourselves and hope that this paper will help other interested scientists to follow-up on and extend as well.

**Recommendations for the authors:**

**Reviewer #1 (Recommendations For The Authors):**
R.1.1: Why these particular traits (body fat, mean corpuscular haemoglobin, neuroticism, heel bone mineral density, monocyte count, sitting height)?

A.8: Traits were initially selected to cover a variety of traits (anthropometric, metabolic, personality..) and to illustrate different examples of tail architecture. However, in response to a point from reviewer 2 (see A.17), we have now overhauled our quality control of traits to ensure that only traits closely matching Gaussian distributions are included. In total, 18 traits were selected, with detailed results presented in Appendix 4 and results corresponding to 6 of the traits presented in the main text (Figure 6) to show examples of different types of tail architecture.

R.1.2: Why are there separate tests for de novo and Mendelian architectures? It seems that one could use either of the derived tests for both purposes, simply by switching to a two-sided test for each tail. My guess is that the score test of whether alpha is zero would be the more statistically powerful test.

A.9: The score test of whether alpha is zero has limited power to detect Mendelian architectures. This is because under Mendelian effects, half the siblings in a family have trait values reflecting the background distribution, such that the mean of sibling trait values is not so different from the polygenic expectation (i.e. alpha close to 0). The Mendelian score test that we developed is substantially more powerful because it evaluates co-occurrence of siblings in the tails, which is far higher under Mendelian architecture in the tail than compared to polygenic architecture.

However, in order test for general departures from our null model, including those of non-Gaussian environmental factors, we now include results from performing a Kolmogorov-Smirnoff test of difference from the expected distribution, and also provide this test as an option in our ‘SibArc’ software tool.

R.1.3: This method assumes that assortative mating is absent. I worry that sitting height might not be a good trait to analyze, since there is some assortative mating (~0.3) for height (e.g., Yengo et al., 2018). Perhaps this trait should not be included among those that are analyzed in this paper. Then again, it is possible that there is less assortative mating for sitting height than total height (i.e., leg length) (Jensen & Sinha, 1993).

A.10: It is true that our method assumes random mating. We note that while assortative mating increases sibling similarity relative to expectation, if it is stable across the trait distribution it will also bias heritability estimation upward which is likely it’s potential impact in our framework. However, if assortative mating is more prevalent in the tails of the distribution, it can result in excess kurtosis – an impact that can increase false positive Mendelian tests and false negative de novo tests. Given that the trait distribution for Sitting Height has only moderate excess Kurtosis (~0.4, see Fig 9, Appendix 4) and we inferred de novo architecture only for this trait, we feel that including it in the paper is appropriate.

R.1.4: I wonder if it's possible to discuss the impact of non-additive genetic variance on the method. How does this affect the estimation of heritability, which calibrates the expectation for regression to the mean? Can non-additive genetic deviations explain a rejection of the null hypothesis of simple polygenicity?

A.11: Yes, the heritability estimation, which calibrates expectation for regression to the mean, assumes additivity of effects, as do the most popular estimators of heritability from GWAS data in the field: GCTA-GREML, LD Score regression and LDAK. Accordingly, non-additive genetic effects could result in rejection of the null hypothesis. We have highlighted this point in the Discussion. However, we also point out that current evidence suggests that the contribution of non-additive genetic effects to complex trait variation is relatively small (Hivert 2021) and that non-additive genetic effects that have a similar impact across the trait distribution should not be a problem for our approach (only those that have an increasing effect towards the tails would be).

R.1.5: p.5: Maybe a more realistic way to simulate a genetic architecture is to draw the MAF from the distribution [MAF(1 - MAF)]^{-1} and then an effect of the minor allele from some mound-shaped distribution (e.g., mixture of normals). The absolute or squared effect of the minor allele should increases as the MAF decreases, and there have been some papers trying to estimate this relationship (e.g., Zeng et al., 2021). Maybe make the number of causal SNPs 10,000. I don't rate this as an urgent suggestion because my sense is that the method should be robust, making adequate even a fairly minimal simulation confirming its accuracy.

A.11: In separate work, we have performed a comprehensive simulation study using the forward-in-time population genetic simulator SLIM-3 (Haller and Messer, 2019), which generates genetic effects according to Gaussian and Gamma distributions and models different selective pressures on complex traits. We plan to publish this work shortly and also extend the simulations to family data, from which we will be able to test the performance of our methods here under a range of different scenarios of genetic variation generation, including a variety of relationships between allele frequency and effect sizes. We agree with the reviewer that at this point, however, our minimal simulation should be sufficient to confirm our tests’ general robustness and so we will perform further testing once we have extended our more sophisticated simulation study.

R.1.6: p.6: Step D seems to leave out a normalization of G to have unit variance. Also, the last part should say "the square of the correlation between the genetic liability and the trait is equal to the heritability."

A.12: Corrected – we thank the reviewer for spotting this.

R.1.7: Figure 5: The power being adequate if roughly 1 of a 1000 index siblings with an extreme trait value owes their values to de novo mutations makes me think that there should be a discussion of the prior probability. The average person carries about 80 de novo mutations. How many of these are likely to affect, e.g., height? Zeng et al. (2021) gave estimates of mutational targets. Given that a mutation affects height, will its likely effect size be large enough to be detected with the method? Kemper et al. (2012) discussed this point in a perhaps useful way.

A.13: We find the work investigating mutational target sizes and generating effect sizes of different mutations (de novo or rare) to be extremely interesting and critical for understanding the causes of observed genetic variation. However, we think that this work is insufficiently progressed at this point to build on directly here for making more nuanced interpretation of our results. We are, however, exploring the impact of mutational target sizes, effect size distributions and selection effects, on the genetic architecture of complex traits via population genetic simulations (see A.11), and so we hope to be able to provide more in-depth interpretation of our results in the future.

R.1.8: Figure 6: The number in the tables for Mendelian architecture are presumably observed and expected counts. But what about the numbers for de novo architecture? Those don't look like counts. Maybe they are conditional expectations of standardized trait values. Whatever the case may be, the caption should provide an explanation.

A.14: The observed and expected values for the de novo statistical test represent the expected and observed mean standardized trait values for siblings of individuals in the bottom and top 1% of the distribution. We have now made this clear in our updated figure.

R.1.9: p. 16: Element (2,1) in the precision matrix after Equation 15 is missing a negative sign.

A.15: Corrected – we thank the reviewer for spotting this.

R.1.10: p. 20: Shouldn't Equation 20 place an exponent of n on the factor outside of the exponential?

A.16: Corrected – we thank the reviewer for spotting this.

**Reviewer #2 (Recommendations For The Authors):**
R.2.1: The first concern that I have is that their statistical tests rely heavily on an assumption of bivariate normal distribution for sibling pair's phenotypes. Real phenotypes do not have such a distribution in general. The authors rely upon an inverse-normal transform when applying their method to real data. While the inverse-normal transform will ensure that the siblings' phenotypes have a marginal normal distribution, such a transform does not ensure that the joint distribution is bivariate normal. The authors should examine their procedure for simulated phenotypes with a non-normal distribution to see if their statistical tests remain properly calibrated. Related to this, I am concerned about applying an inverse normal transform to the neuroticism phenotype that contains only 13 unique values in UKB. How does the transform deal with tied values? Can we sensibly talk about extreme trait values for such a set of observations?

A.17: The reviewer is correct that a bivariate normal distribution for sibling pairs’ trait values does not necessarily hold, and only does so if the assumptions of our null model are met (polygenic effects, Gaussian environmental effects, random mating..). We have now more clearly described the assumptions of our null model, and to increase the matching of our selected traits to those assumptions we have expanded our analyses and now present results on traits that are close to Gaussian. As part of this more strict quality control, only traits with more than 50 unique values are included, meaning that neuroticism is excluded in our final analysis. We also now note that performing an inverse normal transformation on the traits only increases the robustness of the tests to some of our modelling assumptions. In future work we plan to investigate how best to model the conditional sibling distribution under a variety of non-Gaussian environmental effects and different non-random patterns of mating.

R.2.2: The joint sibling phenotype distribution (Equation 4) can be derived by applying the formula for the conditional distribution of a multivariate Gaussian to the standard additive genetic model. The authors' derivation is unnecessarily complex. Furthermore, many of the formulae have been used in Shai Carmi's work on embryo screening, but this work is not cited.

A.18: We now state in the text that the conditional sibling distribution can also be derived from the joint trait distribution of related individuals, which we use in our extension to the 3-sibling scenario, and cite Shai Carmi’s work where this is used. The joint distribution is a more straightforward way to derive the conditional sibling distribution, but our derivation based on considering mid-parents is generalisable to cases where assumptions of random mating, Gaussian population trait distribution and no selection do not hold. We also think that our mid-parent based derivation will be more intuitive to many readers, leading to greater understanding and potential for extension. Therefore, overall we believe that its presentation is worthwhile and we have now elaborated on this in the Methods.

R.2.3: Equation 8: this probability should be conditional on s1

A.19: Corrected – we thank the reviewer for spotting this.

R.2.4: The empirical application to UKB data is lacking methodological details. Also, the number of siblings used is low compared to the number of available sibling pairs. Around 19k sibling pairs are available in the UKB white British subsample, but only 10k were used for height. Why? Also, why are extreme values excluded? Isn't this removing the signal the authors are looking to explain?

A.20: We have now provided more methodological details throughout the Methods section, in particular in relation to the samples used and quality control performed. The removal of individuals with extreme values, in particular, is because unusually low/high trait values are more likely to be due to measurement error (e.g. due to imperfect measuring device, or storage/assaying) than for typical values, and so while this may also result in some loss in power (albeit small due to few individuals having values +/- 8 s.d. trait means) we consider it worth it for the potential reduction in type I error. In performing our newly expanded analysis (described above), and accounting for the reviewer’s point here about sample size, we did find a bug in our pipeline that meant that we did not include as many sibling pairs as available. We thank the reviewer for spotting this, since this contributed to our new analysis being substantially more powerful than the original (including up to ~17k sibling pairs depending on completeness of trait data).

Benjamin C Haller, Phillip W Messer. SLiM 3: Forward Genetic Simulations Beyond the Wright–Fisher Model. Molecular Biology and Evolution. 2019. 36(3): 632-637.

SD Whiteman, SM McHale, A Soli. Theoretical Perspectives on Sibling Relationships. J Fam Theory Rev. 2011 Jun 1;3(2):124-139.

Nicholas H Barton, Alison M Etheridge, and Amandine Véber. The infinitesimal model: Definition, derivation, and implications. Theoretical population biology, 118:50–73, 2017.

Valentin Hivert et al. “Estimation of non-additive genetic variance in human complex traits from a large sample of unrelated individuals.” American journal of human genetics vol. 108,5 (2021)